# Factorizable Graph Convolutional Networks

**Yiding Yang**
Stevens Institute of Technology
yyang99@stevens.edu

**Zunlei Feng**
Zhejiang University
zunleifeng@zju.edu.cn

**Mingli Song**
Zhejiang University
brooksong@zju.edu.cn

**Xinchao Wang** *
Stevens Institute of Technology
xinchao.wang@stevens.edu

## Abstract

Graphs have been widely adopted to denote structural connections between entities. The relations are in many cases heterogeneous, but entangled together and denoted merely as a single edge between a pair of nodes. For example, in a social network graph, users in different latent relationships like friends and colleagues, are usually connected via a bare edge that conceals such intrinsic connections. In this paper, we introduce a novel graph convolutional network (GCN), termed as *factorizable graph convolutional network* (FactorGCN), that explicitly disentangles such intertwined relations encoded in a graph. FactorGCN takes a simple graph as input, and disentangles it into several factorized graphs, each of which represents a latent and disentangled relation among nodes. The features of the nodes are then aggregated separately in each factorized latent space to produce disentangled features, which further leads to better performances for downstream tasks. We evaluate the proposed FactorGCN both qualitatively and quantitatively on the synthetic and real-world datasets, and demonstrate that it yields truly encouraging results in terms of both disentangling and feature aggregation. Code is publicly available at https://github.com/ihollywhy/FactorGCN.PyTorch.

## 1 Introduction

Disentangling aims to factorize an entity, like a feature vector, into several interpretable components, so that the behavior of a learning model can be better understood. In recent years, many approaches have been proposed towards tackling disentangling in deep neural networks and have achieved promising results. Most prior efforts, however, have been focused on the disentanglement of convolutional neural network (CNN) especially the auto-encoder architecture, where disentangling takes place during the stage of latent feature generation. For example, VAE [1] restrains the distribution of the latent features to Gaussian and generates disentangled representation; $\beta$-VAE [2] further improves the disentangling by introducing $\beta$ to balance the independence constraints and reconstruction accuracy.

Despite the many prior efforts in CNN disentangling, there are few endeavors toward disentangling in the irregular structural domain, where graph convolutional network (GCN) models are applied. Meanwhile, the inherent differences between grid-like data and structural data precludes applying CNN-based disentangling methods to GCN ones. The works of [3, 4], as pioneering attempts, focus on the node-level neighbour partition and ignore the latent multi-relations among nodes.

We introduce in this paper a novel GCN, that aims to explicitly conduct graph-level disentangling, based on which convolutional features are aggregated. Our approach, termed as *factorizable graph*

---

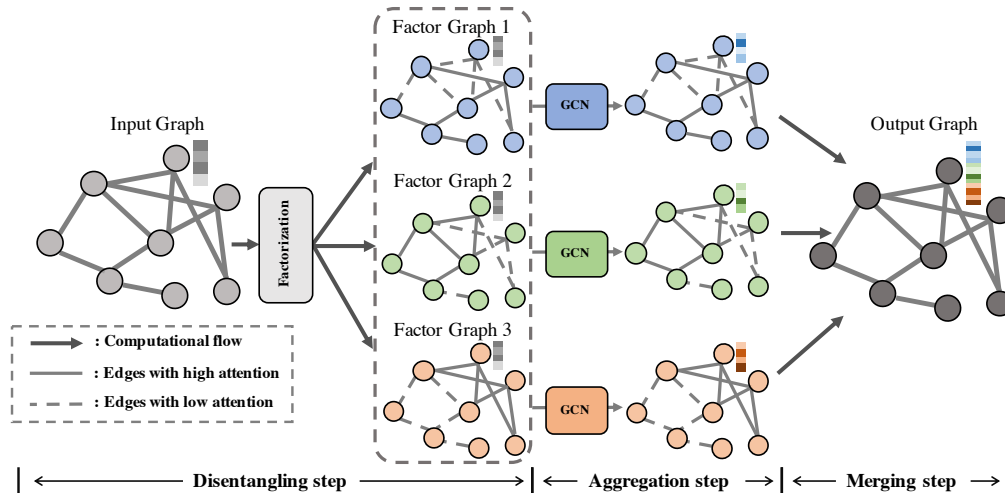

Figure 1: Illustration of one layer in the proposed FactorGCN. It contains three steps: *Disentangling*, *Aggregation*, and *Merging*. In the disentangling step, the input graph is decomposed into several factor graphs, each of which represents a latent relation among nodes. In the aggregation step, GCNs are applied separately to the derived factor graphs and produce the latent features. In the merging step, features from all latent graphs are concatenated to form the final features, which are block-wise interpretable.

*convolutional network* (FactorGCN), takes as input a *simple graph*, and decomposes it into several factor graphs, each of which corresponds to a disentangled and interpretable relation space, as shown in Fig. 1. Each such graph then undergoes a GCN, tailored to aggregate features only from one disentangled latent space, followed by a merging operation that concatenates all derived features from disentangled spaces, so as to produce the final block-wise interpretable features. These steps constitute one layer of the proposed FactorGCN. As the output graph with updated features share the identical topology as input, nothing prevents us from stacking a number of layers to disentangle the input data at different levels, yielding a hierarchical disentanglement with various numbers of factor graph at different levels.

FactorGCN, therefore, potentially finds application in a wide spectrum of scenarios. In many real-world graphs, multiple heterogeneous relations between nodes are mixed and collapsed to one single edge. In the case of social networks, two people may be *friends*, *colleagues*, and *living in the same city* simultaneously, but linked via one single edge that omits such interconnections; in the co-purchasing scenario [5], products are bought together for different reasons like *promotion*, and *functional complementary*, but are often ignored in the graph construction. FactorGCN would, in these cases, deliver a disentangled and interpretable solution towards explaining the underlying rationale, and provide discriminant learned features for the target task.

Specifically, the contributions of FactorGCN are summarized as follows.

- **Graph-level Disentangling**. FactorGCN conducts disentangling and produces block-wise interpretable node features by analyzing the whole graph all at once, during which process the global-level topological semantics, such as the higher-order relations between edges and nodes, is explicitly accounted for. The disentangled factor graphs reveal latent-relation specific interconnections between the entities of interests, and yield interpretable features that benefit the downstream tasks. This scheme therefore contrasts to the prior approaches of [3, 4], where the disentanglement takes place only within a local neighborhood, without accounting for global contexts.

- **Multi-relation Disentangling**. Unlike prior methods that decode only a single attribute for a neighboring node, FactorGCN enables multi-relation disentangling, meaning that the center node may aggregate information from a neighbour under multiple types of relations. This mechanism is crucial since real-world data may contain various relations among the same pair of entities. In the case of a social network graph, for example, FactorGCN would

produce disentangled results allowing for two users to be both *friends* and *living in the same city*; such multi-relation disentangling is not supported by prior GCN methods.

- **Quantitative Evaluation Metric**. Existing quantitative evaluation methods [6, 7] in the grid domain rely on generative models, like auto-encoder [8] or GAN [9]. Yet in the irregular domain, unfortunately, state-of-the-art graph generative models are only applicable for generating small graphs or larger ones without features. Moreover, these models comprise a sequential generation step, making it infeasible to be integrated into the graph disentangling frameworks. To this end, we propose a graph edit-distance based metric, which bypasses the generation step and estimates the similarity between the factor graphs and the ground truth.

We conducted experiments on five datasets in various domains, and demonstrate that the proposed FactorGCN yields state-of-the-art performances for both disentanglement and downstream tasks. This indicates that, even putting side its disentangling capability, FactorGCN may well serve as a general GCN framework. Specifically, on the ZINC dataset [10], FactorGCN outperforms other methods by a large margin, and, without the bond information of the edges, FactorGCN achieves a performance on par with the state-of-the-art method that explicitly utilizes edge-type information.

## 2   Related Work

**Disentangled representation learning**. Learning disentangled representations has recently emerged as a significant task towards interpretable AI [11, 12]. Unlike earlier attempts that rely on handcrafted disentangled representations or variables [13, 14], most of the recent works in disentangled representation learning are based on the architecture of auto-encoder [2, 15, 16, 7, 17, 8] or generative model [9, 18, 19]. One mainstream auto-encoder approach is to constrain the latent feature generated from the encoder to make it independent in each dimension. For example, VAE [1] constrains the distribution of the latent features to Gaussian; $\beta$-VAE[2] enlarges the weight of the KL divergence term to balance the independence constraints and reconstruction accuracy; [20] disentangles the latent features by ensuring that each block of latent features cannot be predicted from the rest; DSD [15] swaps some of the latent features twice to achieve semi-supervised disentanglement. For the generative model, extra information is introduced during the generation. For example, InfoGAN [9] adds the class code to the model and maximizes the mutual information between the generated data and the class code.

**Graph convolutional network**. Graph convolutional network (GCN) has shown its potential in the non-grid domain [21–26], achieving promising results on various type of structural data, like citation graph [27], social graph [28], and relational graph [29]. Besides designing GCN to better extract information from non-grid data, there are also a couple of works that explore the disentangled GCNs [30, 4]. DisenGCN [3] adopts neighbour routine to divide the neighbours of the node into several mutually exclusive parts. IPGDN [4] improves DisenGCN by making the different parts of the embedded feature independent. Despite results of the previous works, there remain still several problems: the disentanglement is in the node level, which does not consider the information of the whole graph, and there is no quantitative metrics to evaluate the performance of disentanglement.

## 3   Method

In this section, we will give a detailed description about the architecture of FactorGCN, whose basic component is the disentangle layer, as shown in Fig. 1.

### 3.1   Disentangling Step

The goal of this step is to factorize the input graph into several factor graphs. To this end, we treat the edges equally across the whole graph. The mechanism we adopt to generate these factorized coefficient is similar to that of graph attention network [27]. We denote the input of the disentangle layer as $\mathbf{h} = \{h_0, h_1, ..., h_n\}, h_i \in \mathcal{R}^F$ and $\mathbf{e} = \{e_0, e_1, ..., e_m\}, e_k = (h_i, h_j)$. $\mathbf{h}$ denotes the set of nodes with feature of $F$ dimension, and $\mathbf{e}$ denotes the set of edges.

The input nodes are transformed to a new space, done by multiplying the features of nodes with a linear transformation matrix $\mathbf{W} \in \mathcal{R}^{F' \times F}$. This is a standard operation in most GCN models, which

increases the capacity of the model. The transformed features are then used to generate the factor coefficients as follows

$$E_{ije} = 1/\left(1 + e^{-\Psi_e(h'_i, h'_j)}\right); h' = \mathbf{W}h, \tag{1}$$

where $\Psi_e$ is the function that takes the features of node $i$ and node $j$ as input and computes the attention score of the edge for factor graph $e$, and takes the form of an one-layer MLP in our implementation; $E_{ije}$ then can be obtained by normalizing the attention score to $[0, 1]$, representing the coefficient of edge from node $i$ to node $j$ in the factor graph $e$; $h'$ is the transformed node feature, shared across all functions $\Psi_*$. Different from most previous forms of attention-based GCNs that normalize the attention coefficients among all the neighbours of nodes, our proposed model generates these coefficients directly as the factor graph.

Once all the coefficients are computed, a factor graph $e$ can be represented by its own $E_e$, which will be used for the next aggregation step. However, without any other constrain, some of the generated factor graphs may contain a similar structure, degrading the disentanglement performance and capacity of the model. We therefore introduce an additional head in the disentangle layer, aiming to avoid the degradation of the generated factor graphs.

The motivation of the additional head is that, a well disentangled factor graph should have enough information to be distinguished from the rest, only based on its structure. Obtaining the solution that all the disentangled factor graphs differ from each other to the maximal degree, unfortunately, is not trivial. We thus approximate the solution by giving unique labels to the factor graphs and optimizing the factor graphs as a graph classification problem. Our additional head will serve as a discriminator, shown in Eq. 2, to distinguish which label a given graph has:

$$G_e = \text{Softmax}\left(f\Big(\text{Readout}(\mathcal{A}(\mathbf{E}_e, h'))\Big)\right). \tag{2}$$

The discriminator contains a three-layer graph auto-encoder $\mathcal{A}$, which takes the transformed feature $h'$ and the generated attention coefficients of factor graph $\mathbf{E}_e$ as inputs, and generates the new node features. These features are then readout to generate the representation of the whole factor graph. Next, the feature vectors will be sent to a classifier with one fully connected layer. Note that all the factor graphs share the same node features, making sure that the information discovered by the discriminator only comes from the difference among the structure of the factor graphs. More details about the discriminator architecture can be found in the supplementary materials.

The loss used to train the discriminator is taken as follows:

$$\mathcal{L}_d = -\frac{1}{N}\sum_i^N \left(\sum_{c=1}^{N_e} \mathbb{1}_{e=c} log(G_i^e[c])\right), \tag{3}$$

where $N$ is the number of training samples, set to be the number of input graphs multiplies by the number of factor graphs; $N_e$ is the number of factor graphs; $G_i^e$ is the distribution of sample $i$ and $G_i^e[c]$ represents the probability that the generated factor graph has label $c$. $\mathbb{1}_{e=c}$ is an indicator function, taken to be one when the predicted label is correct.

## 3.2  Aggregation Step

As the factor graphs derived from the disentangling step is optimized to be as diverse as possible, in the aggregation step, we will use the generated factor graphs to aggregate information in different structural spaces.

This step is similar as the most GCN models, where the new node feature is generated by taking the weighted sum of its neighbors. Our aggregation mechanism is based on the simplest one, which is used in GCN [28]. The only difference is that the aggregation will take place independently for each of the factor graphs.

The aggregation process is formulated as

$$h_i^{(l+1)e} = \sigma(\sum_{j \in \mathcal{N}_i} E_{ije}/c_{ij}h_j^{(l)}\mathbf{W}^{(l)}), c_{ij} = (|\mathcal{N}_i||\mathcal{N}_j|)^{1/2}, \tag{4}$$

where $h_i^{(l+1)e}$ represents the new feature for node $i$ in $l+1$ layer aggregated from the factor graph $e$; $\mathcal{N}_i$ represents all the neighbours of node $i$ in the input graph; $E_{ije}$ is the coefficient of the edge from

node $i$ to node $j$ in the factor graph $e$; $c_{ij}$ is the normalization term that is computed according to the degree of node $i$ and node $j$; $\mathbf{W}^{(l)}$ is a linear transformation matrix, which is the same as the matrix used in the disentangling step.

Note that although we use all the neighbours of a node in the input graph to aggregate information, some of them are making no contribution if the corresponding coefficient in the factor graph is zero.

## 3.3 Merging Step

Once the aggregation step is complete, different factor graphs will lead to different features of nodes. We merge these features generated from different factor graphs by applying

$$h_i^{(l+1)} = ||_{e=1}^{N_e} h_i^{(l+1)e},\tag{5}$$

where $h_i^{(l+1)}$ is the output feature of node $i$; $N_e$ is the number of factor graphs; $||$ represents the concatenation operation.

## 3.4 Architecture

We discuss above the design of one disentangle layer, which contains three steps. The FactorGCN model we used in the experimental section contains several such disentangle layers, increasing the power of expression. Moreover, by setting different number of factor graphs in different layers, the proposed model can disentangle the input data in a hierarchical manner.

The total loss to train FactorGCN model is $\mathcal{L} = \mathcal{L}_t + \lambda * \mathcal{L}_d$. $\mathcal{L}_t$ is the loss of the original task, which is taken to be a binary cross entropy loss for multi-label classification task, cross entropy loss for multi-class classification task, or L1 loss for regression task. $\mathcal{L}_d$ is the loss of the discriminator we mentioned above. $\lambda$ is the weight to balance these two losses.

# 4 Experiments

In this section, we show the effectiveness of the proposed FactorGCN, and provide discussions on its various components as well as the sensitivity with respect to the key hyper-parameters. More results can be found in the supplementary materials.

## 4.1 Experimental setups

**Datasets**. Here, we use six datasets to evaluate the effectiveness of the proposed method. The first one is a synthetic dataset that contains a fixed number of predefined graphs as factor graphs. The second one is the ZINC dataset [31] built from molecular graphs. The third one is Pattern dataset [31], which is a large scale dataset for node classification task. The other three are widely used graph classification datasets include social networks (COLLAB,IMDB-B) and bioinformatics graph (MUTAG) [32]. To generate the synthetic dataset that contains $N_e$ factor graphs, we first generate $N_e$ predefined graphs, which are the well-known graphs like Turán graph, house-x graph, and balanced-tree graph. We then choose half of them and pad them with isolated nodes to make the number of nodes to be 15. The padded graphs will be merged together as a training sample. The label of the synthetic data is a binary vector, with the dimension $N_e$. Half of the labels will be set to one according to the types of graphs that the sample generated from, and the rest are set to zero. More information about the datasets can be found in the supplemental materials.

**Baselines**. We adopt several methods, including state-of-the-art ones, as the baselines. Among all, MLP is the simplest one, which contains multiple fully connected layers. Although this method is simple, it can in fact perform well when comparing with other methods that consider the structural information. We use MLP to check whether the other compared methods benefit from using the structural information as well. GCN aggregates the information in the graph according to the laplacian matrix of the graph, which can be seen as a fixed weighted sum on the neighbours of a node. GAT [27] extends the idea of GCN by introducing the attention mechanism. The weights when doing the aggregation is computed dynamically according to all the neighbours. For the ZINC dataset, we also add MoNet [25] and GatedGCN$_E$ [31] as baselines. The former one is the state-of-the-art method that does not use the type information of edges while the latter one is the state-of-the-art one that uses additional edge information. Random method is also added to provide the result of

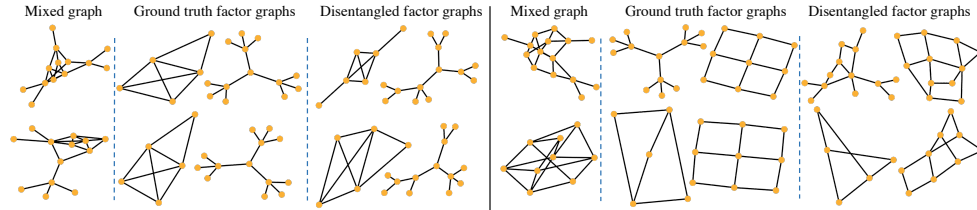

| Mixed graph | Ground truth factor graphs | Disentangled factor graphs | Mixed graph | Ground truth factor graphs | Disentangled factor graphs |

Figure 2: Examples of the disentangled factor graphs on the synthetic dataset. The isolated nodes are eliminated for a better visualization.

random guess for reference. For the other three graph datasets, we add non DL-based methods (WL subtree, PATCHYSAN, AWL) and DL-based methods (GCN, GraphSage [33], GIN) as baselines. DisenGCN [3] and IPDGN [4] are also added.

**Hyper-parameters**. For the synthetic dataset, Adam optimizer is used with a learning rate of 0.005, the number of training epochs is set to 80, the weight decay is set to 5e-5. The row of the adjacent matrix of the generated synthetic graph is used as the feature of nodes. The negative slope of LeakyReLU for GAT model is set to 0.2, which is the same as the original setting. The number of hidden layers for all models is set to two. The dimension of the hidden feature is set to 32 when the number of factor graphs is no more than four and 64 otherwise. The weight for the loss of discriminator in FactorGCN is set to 0.5.

For the molecular dataset, the dimension of the hidden feature is set to 144 for all methods and the number of layers is set to four. Adam optimizer is used with a learning rate of 0.002. No weight decay is used. $\lambda$ of FactorGCN is set to 0.2. All the methods are trained for 500 epochs. The test results are obtained using the model with the best performance on validation set. For the other three datasets, three layers FactorGCN is used.

## 4.2 Qualitative Evaluation

We first provide the qualitative evaluations of disentanglement performance, including the visualization of the disentangled factor graphs and the correlation analysis of the latent features.

**Visualization of disentangled factor graphs**. To give an intuitive understanding of the disentanglement. We provide in Fig. 2 some examples of the generated factor graphs. We remove the isolated nodes and visualize the best-matched factor graphs with ground truths. More results and analyses can be found in the supplemental materials.

**Correlation of disentangled features**. Fig. 3 shows the correlation analysis of the latent features obtained from several pre-trained models on the synthetic dataset. It can be seen that also GCN and MLP models can achieve a high performance in the downstream task, and their latent features are hidden entangled. GAT gives more independent latent features but the performance is degraded in the original task. FactorGCN is able to extract the highly independent latent features and meanwhile achieve a better performance in the downstream task.

## 4.3 Quantitative Evaluation

The quantitative evaluation focuses on two parts, the performance of the downstream tasks and that of the disentanglement.

**Evaluation protocol**. For the downstream tasks, we adopt the corresponding metrics to evaluate, i.e., Micro-F1 for the multi-label classification task, mean absolute error (MAE) for the regression task. We design two new metrics to evaluate the disentanglement performance on the graph data. The first one is graph edit distance on edge ($\text{GED}_E$). This metric is inspired by the traditional graph edit distance (GED). Since the input graph already provides the information about the order of nodes, the disentanglement of the input data, in reality, only involves the changing of edges. Therefore, we restrict the GED by only allowing adding and removing the edges, and thus obtain a score of $\text{GED}_E$ by Hungarian match between the generated factor graphs and the ground truth.

Specifically, for each pair of the generated factor graph and the ground truth graph, we first convert the continuous value in the factor graph to 1/0 value by setting the threshold to make the number of

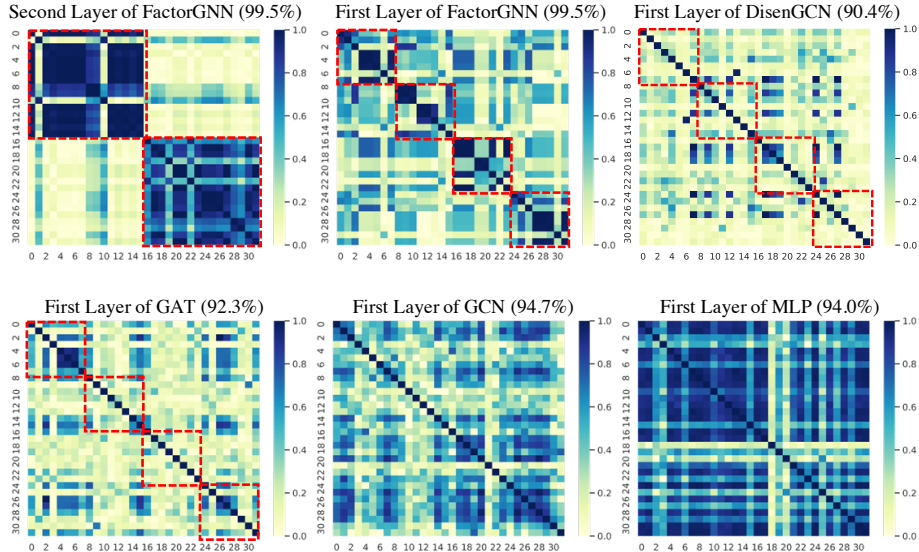

Figure 3: Feature correlation analysis. The hidden features are obtained from the test split using the pre-trained models on the synthetic dataset. It can be seen that the features generated from FactorGCN present a more block-wise correlation pattern, indicating that the latent features have indeed been disentangled. We also show the classification performance in brackets.

Table 1: Performance on synthetic dataset. The four methods are evaluated in terms of the classification and the disentanglement performance. Classification performance is evaluated by Micro-F1 and disentanglement performance is measured by $GED_E$ and C-Score. For each method, we run the experiments five times and report the mean and std. Random method generates four factor graphs. GAT_W/Dis represents GAT model with the additional discriminator proposed in this paper.

|  | MLP | GCN | GAT | GAT_W/Dis | DisenGCN | FactorGCN (Ours) | Random |
|---|---|---|---|---|---|---|---|
| Micro-F1 ↑ | $0.940 \pm 0.002$ | $0.947 \pm 0.003$ | $0.923 \pm 0.009$ | $0.928 \pm 0.009$ | $0.904 \pm 0.007$ | $\mathbf{0.995 \pm 0.004}$ | $0.250 \pm 0.002$ |
| $GED_E$ ↓ | - | - | $12.59 \pm 3.00$ | $12.35 \pm 3.86$ | $\mathbf{10.54 \pm 4.35}$ | $10.59 \pm 4.37$ | $32.09 \pm 4.85$ |
| C-Score ↑ | - | - | $0.288 \pm 0.064$ | $0.274 \pm 0.065$ | $0.367 \pm 0.026$ | $\mathbf{0.532 \pm 0.044}$ | $0.315 \pm 0.002$ |

edges in these two graphs are the same. Then, $GED_E$s can be computed for every such combination. Finally, Hungarian match is adopted to obtain the best bipartite matching results as the $GED_E$ score.

Besides the $GED_E$ score, we also care about the consistency of the generated factor graph. In other words, the best-matched pairs between the generated factor graphs and the ground truths, optimally, should be identical across all samples. We therefore introduce the second metric named as consistency score (C-Score), related to $GED_E$. C-Score is computed as the average percentage of the most frequently matched factor graphs. The C-score will be one if the ground truth graphs are always matched to the fixed factor graphs. A more detailed description of evaluation protocol can be found in the supplemental materials.

**Evaluation on the synthetic dataset**. We first evaluate the disentanglement performance on a synthetic dataset. The results are shown in Tab. 1. Although MLP and GCN achieve good classification performances, they are not capable of disentanglement. GAT disentangles the input by using multi-head attention, but the performance of the original task is degraded. Our proposed method, on the other hand, achieves a much better performance in terms of both disentanglement and the original task. We also evaluate the compared methods on the synthetic dataset with various numbers of factor graphs, shown in Tab. 2. As the number of latent factor graphs increase, the performance gain of the FactorGCN becomes large. However, when the number of factor graphs becomes too large, the task will be more challenging, yielding lower performance gains.

**Evaluation on the ZINC dataset**. For this dataset, the type information of edges is hidden during the training process, and is serve as the ground truth to evaluate the performance of disentanglement. Tab. 3 shows the results. The proposed method achieves the best performance on both the disentan-

Table 2: Classification performance on synthetic graphs with different numbers of factor graphs. We change the total number of factor graphs and generate five synthetic datasets. When the number of factor graphs increases, the performance gain of FactorGCN becomes larger. However, as the number of factor graphs becomes too large, disentanglement will be more challenging, yielding lower performance gains.

| Method | Number of factor graphs | | | | |
|---|---|---|---|---|---|
| | 2 | 3 | 4 | 5 | 6 |
| MLP | $1.000 \pm 0.000$ | $0.985 \pm 0.002$ | $0.940 \pm 0.002$ | $0.866 \pm 0.001$ | $0.809 \pm 0.002$ |
| GCN | $1.000 \pm 0.000$ | $0.984 \pm 0.000$ | $0.947 \pm 0.003$ | $0.844 \pm 0.002$ | $0.765 \pm 0.001$ |
| GAT | $1.000 \pm 0.000$ | $0.975 \pm 0.002$ | $0.923 \pm 0.009$ | $0.845 \pm 0.006$ | $0.791 \pm 0.006$ |
| FactorGCN | $1.000 \pm 0.000$ | $\mathbf{1.000 \pm 0.000}$ | $\mathbf{0.995 \pm 0.004}$ | $\mathbf{0.893 \pm 0.021}$ | $\mathbf{0.813 \pm 0.049}$ |

Table 3: Performance on the ZINC dataset. FactorGCN outperforms the compared methods by a large margin, with the capability of disentanglement. Note that our proposed method even achieves a similar performance as GatedGCN$_E$, the state-of-the-art method on ZINC dataset that explicitly uses additional edge information.

| | MLP | GCN | GAT | MoNet | DisenGCN | FactorGCN (Ours) | GatedGCN$_E$ |
|---|---|---|---|---|---|---|---|
| MAE ↓ | $0.667 \pm 0.002$ | $0.503 \pm 0.005$ | $0.479 \pm 0.010$ | $0.407 \pm 0.007$ | $0.538 \pm 0.005$ | $\mathbf{0.366 \pm 0.014}$ | $0.363 \pm 0.009$ |
| GED$_E$ ↓ | - | - | $15.46 \pm 6.06$ | - | $14.14 \pm 6.19$ | $\mathbf{12.72 \pm 5.34}$ | - |
| C-Score ↑ | - | - | $0.309 \pm 0.013$ | - | $0.342 \pm 0.034$ | $\mathbf{0.441 \pm 0.012}$ | - |

glement and the downstream task. We also show the state-of-the-art method GatedGCN$_E$ on this dataset on the right side of Tab. 3, which utilizes the type information of edges during the training process. Our proposed method, without any additional edge information, achieves truly promising results that are to that of GatedGCN$_E$, which needs the bond information of edges during training.

**Evaluation on more datasets**. To provide a thorough understanding of the proposed method, We also carry out evaluations on three widely used graph classification datasets and one node classification dataset to see the performances of FactorGCN as a general GCN framework. The same 10-fold evaluation protocol as [21] is adopted. Since there are no ground truth factor graphs, we only report the accuracy, shown in Tab. 4 and Tab. 5. Our method achieves consistently the best performance, showing the potential of the FactorGCN as a general GCN framework, even putting aside its disentangling capability. More details about the evaluation protocol, the setup of our method, and the statistic information about these datasets can be found in the supplemental materials.

## 4.4 Ablation and sensitivity analysis

We show in Fig. 4 the ablation study and sensitivity analysis of the proposed method. When varying $\lambda$, the number of factors is set to be eight; when varying the number of factors, $\lambda$ is set to be 0.2. As can be seen from the left figure, the performance of both the disentanglement and the downstream task will degrade without the discriminator. The right figure shows the relations between the performance and the number of factor graphs we used in FactorGCN. Setting the number of factor graphs to be slightly larger than that of the ground truth, in practice, leads to a better performance.

Table 4: Accuracy (%) on three graph classification datasets. FactorGCN performances on par with or better than the state-of-the-art GCN models. We highlight the best DL-based methods and non DL-based methods separately. FactorGCN uses the same hyper-parameters for all datasets.

| | WL subtree | PATCHYSAN | AWL | GCN | GraphSage | GIN | FactorGCN |
|---|---|---|---|---|---|---|---|
| IMDB-B | $73.8 \pm 3.9$ | $71.0 \pm 2.2$ | $\mathbf{74.5 \pm 5.9}$ | $74.0 \pm 3.4$ | $72.3 \pm 5.3$ | $75.1 \pm 5.1$ | $\mathbf{75.3 \pm 2.7}$ |
| COLLAB | $\mathbf{78.9 \pm 1.9}$ | $72.6 \pm 2.2$ | $73.9 \pm 1.9$ | $79.0 \pm 1.8$ | $63.9 \pm 7.7$ | $80.2 \pm 1.9$ | $\mathbf{81.2 \pm 1.4}$ |
| MUTAG | $90.4 \pm 5.7$ | $\mathbf{92.6 \pm 4.2}$ | $87.9 \pm 9.8$ | $85.6 \pm 5.8$ | $77.7 \pm 1.5$ | $\mathbf{89.4 \pm 5.6}$ | $89.9 \pm 6.5$ |

Table 5: Accuracy (%) on the Pattern dataset for node-classification task. FactorGCN achieves the best performance, showing its ability to serve as a general GCN framework.

| GCN | GatedGCN | GIN | MoNet | DisenGCN | IPDGN | FactorGCN |
|---|---|---|---|---|---|---|
| $63.88 \pm 0.07$ | $84.48 \pm 0.12$ | $85.59 \pm 0.01$ | $85.48 \pm 0.04$ | $75.01 \pm 0.15$ | $78.70 \pm 0.11$ | $\mathbf{86.57 \pm 0.02}$ |

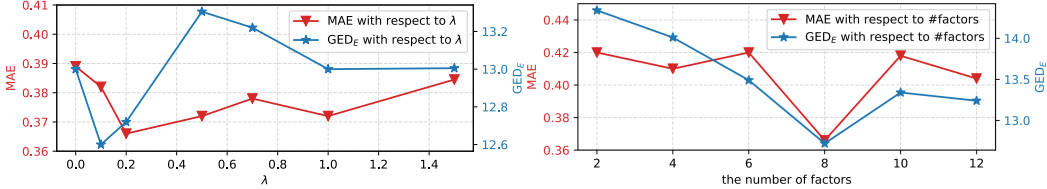

Figure 4: The influence of the balanced weight $\lambda$ and the number of factor graphs.

## 5 Conclusion

We propose a novel GCN framework, termed as FactorGCN, which achieves graph convolution through graph-level disentangling. Given an input graph, FactorGCN decomposes it into several interpretable factor graphs, each of which denotes an underlying interconnections between entities, and then carries out topology-aware convolutions on each such factor graph to produce the final node features. The node features, derived under the explicit disentangling, are therefore block-wise explainable and beneficial to the downstream tasks. Specifically, FactorGCN enables multi-relation disentangling, allowing information propagation between two nodes to take places in disjoint spaces. We also introduce two new metrics to measure the graph disentanglement performance quantitatively. FactorGCN outperforms other methods on both the disentanglement and the downstream tasks, indicating the proposed method is ready to serve as a general GCN framework with the capability of graph-level disentanglement.

## Acknowledgments

This work is supported by the startup funding of Stevens Institute of Technology.

## Broader Impact

In this work we introduce a GCN framework, termed as FactorGCN, that explicitly accounts for disentanglement FactorGCN is applicable to various scenarios, both technical and social. For conventional graph-related tasks, like node classification of the social network and graph classification of the molecular graph, our proposed method can serve as a general GCN framework. For disentangling tasks, our method generates factor graphs that reveal the latent relations among entities, and facilitate the further decision making process like recommendation. Furthermore, given sufficient data, FactorGCN can be used as a tool to analyze social issues like discovering the reasons for the quick spread of the epidemic disease in some areas. Like all learning-based methods, FactorGCN is not free of errors. If the produced disentangled factor graphs are incorrect, for example, the subsequent inference and prediction results will be downgraded, possibly yielding undesirable bias.

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
