[Supplementary Material]

# Factorizable Graph Convolutional Networks
## – *Supplementary Material* –

**Yiding Yang**
Stevens Institute of Technology
yyang99@stevens.edu

**Zunlei Feng**
Zhejiang University
zunleifeng@zju.edu.cn

**Mingli Song**
Zhejiang University
brooksong@zju.edu.cn

**Xinchao Wang** *
Stevens Institute of Technology
xinchao.wang@stevens.edu

In this document we provide supplementary materials that we are not able to fit into the main manuscript due to the page limit. Specifically, we show the architecture of the discriminator, dataset information, setups on all datasets, evaluation protocals, and more visualizations of the disentangled factor graphs.

## A. Discriminator Architecture

The discriminator is taken to be a three-layer GCN, followed by a one-layer MLP. The input of the discriminator is the transformed features of nodes with dimmension $F$, and the factor graphs. The dimensions of the hidden features of the three-layer GCN are set to $F$, $F/2$, and $F$ respectively. The non-linear activation function is Tanh and no residual connection is used. Mean pooling is then applied to the generated node features to obtain the features of graphs. These features are then fed to the one-layer MLP to generate the activation for each class.

## B. Dataset Information

We list the information of all datasets used in the manuscript in Tab. 1.

**The synthetic dataset** contains 20,000 samples. 14,000 for training, 2,000 for validation, and 4,000 for testing. The number of nodes is fixed to 15 and the average degree of samples is 3.43. The task for this dataset is multi-label graph classification, where the number of labels is the same as the number of factor graphs.

**ZINC** dataset contains 12,000 samples. The task of this dataset is to regress a molecular property (the constrained solubility) [Jin et al., 2018]. The number of nodes per graph ranges from 9 to 37, and the average degree of samples is 2.15. Since there are three different type of bonds in the graphs, the number of factor graphs to disentangle is taken to be three.

**The other three datasets** are widely used graph classification datasets. IMDB-B is a movie collaboration dataset. Each graph is derived from a pre-specified genre of movies, where nodes represent actors/actresses and edges represent whether two actors/actresses appear in a same movie. The task is to predict the genre of graphs. COLLAB is a scientific collaboration dataset generated from three public collaboration datasets: High Energy Physics, Condensed Matter Physics, and Astro Physics. Each graph represents a researcher and the task is to predict the field of this researcher. MUTAG is a dataset of 188 mutagenic aromatic and heteroaromatic nitro compounds with 7 discrete labels.

---

Table 1: Dataset information used in the main manuscript include the data split, statistic information, and the tasks.

| Dataset | Dataset information | | | | |
|---|---|---|---|---|---|
| | Split (train/val/test) | #Node per graph | Average degree | #Factor graphs | Task |
| Synthetic | 14,000/2,000/4,000 | 15 | 3.42 | 4 | multi-label graph classification |
| ZINC | 10,000/1,000/1,000 | 9-37 | 2.15 | 3 | graph regression |
| IMDB-B | 10-fold (total 1000) | 19.8 | 10.78 | - | multi-class graph classification (2 classes) |
| COLLAB | 10-fold (total 5000) | 74.5 | 67.61 | - | multi-class graph classification (3 classes) |
| MUTAG | 10-fold (total 188) | 17.9 | 3.20 | - | multi-class graph classification (2 classes) |

## C. Setups on All Datasets

For the synthetic dataset, two disentangle layers with eight and four factor graphs respectively are used for FactorGCN models. The output is then mean pooled followed by a fully connected layer. For the ZINC dataset, four disentangle layers with eight, eight, four, two factor graphs respectively are used. Batch normalization is used after each layer and the output is mean pooled followed by two fully connected layers. For the other three datasets, four disentangle layers with four, four, two, and two factor graphs respectively are used. Batch normalization is used after each layer and the output of each layer are followed by a fully connected layer. All outputs of the fully connected layers are then summed up to compute the final output.

For the three graph classification datasets, we use the same ten-fold cross-validation training strategy as [Xu et al., 2018]. The dataset is separated into ten parts. We generate ten validation accuracy curves when setting each of parts as the validation one. The ten curves are then averaged. The maximum accuracy of the averaged curves and the corresponding standard deviation are reported.

## D. Evaluation Protocols

For the downstream tasks, the evaluation protocols are the same as the original ones, e.g., Micro-F1 for multi-label classification takes, Accuracy for multi-class classification tasks, and MAE for regression tasks.

Two new metrics are designed in this work named $GED_E$ and C-score. We will provide a more detailed description and examples for them. $GED_E$ is based on the definition of graph edit distance (GED), GED measures distance between two graphs. The distance is represented by the steps used to convert one graph to the other, where each of the steps can only be *vertex insertion*, *vertex deletion*, *edge insertion*, or *edge deletion*. However, computing graph edit distance is NP-complete [Gao et al., 2010]. Although we have some approximation algorithm to compute GED [Abu-Aisheh et al., 2015], the computational complexity is still high, making it infeasible to evaluate the disentanglement performance handily.

Since the order of nodes is in fact known through the input graph, we do not need to consider the *vertex insertion* and *vertex deletion* operators during the computation of GED. The GED, for which we eliminate two operations, is named as $GED_E$. Computing $GED_E$ can be very fast since we can just add up the two adjacent matrices, which contain only zeros and ones, and count the number of ones in the new matrix. For each pair of ground truth graphs and disentangled factor graphs, we can compute the best match through Hungarian algorithm and obtain the minimum cost. The $GED_E$ metric is then computed as the average cost together with the standard deviation of the costs.

The $GED_E$ reflects that the capability of the method to generate the latent factor graphs. We also care about the stability of the disentanglement, meaning that we want a specific factor graph that always represents a specific ground truth factor graph and remains the same across different input samples. We thus introduce the consistency score (C-score) to measure the stability of the disentanglement. C-score is computed based on the result of the Hungarian match. For each sample, the Hungarian match result will give us the optimal assignment. After matching all the samples, we will obtain a matched list for each type of the ground truth factor graphs, i.e. four lists if we have four types of factor graphs. For each list, we count the most frequently generated factor graphs and compute the percentage of it in the whole list. Averaging the results from all lists gives us the C-score.

# F. More Visualization of Disentangled Factor Graphs

We provide in Fig. 1 more visualizations of the samples from the synthetic and the disentangled factor graphs. The visualized disentangled factor graphs are the best matched factor graphs, after we set the threshold of the coefficient to make the edges in the factor graph to be the same as the ground truth one.

Figure 1: More visualizations of the mixed graph, ground truth factor graphs, and the disentangled factor graphs on the synthetic dataset.