[Reviews · NeurIPS 2020]

Review 1

Summary and Contributions: The authors of this submission propose a GCN-based method to conduct disentanglement in the graph domain. The proposed method disentangles the input graph into several factor graphs in a graph level, and allows for multiple relations between a pair of entities. This is done by the graph-level attentional aggregation strategy during the training process. A new metric is designed to evaluate the disentanglement performance quantitatively. The proposed method achieves the best disentanglement performance and very competitive performance on the downstream tasks even without the additional edge information.

Strengths: 1)This paper is well written. The proposed method is well motivated and the description is clear. 2)Besides the qualitative evaluation used in the previous methods, this paper provides a legitimate and practical quantitative evaluation protocol to evaluate the performance of the disentanglement in the graph domain, which could be of interest to a large audience. 3)The experimental results on the ZINC dataset are impressive. The proposed method achieves very similar performance as the model that uses the edge information. Such a result aligns well with the claim that the disentangled factor graphs can further boost the performance of the downstream tasks.

Weaknesses: 1) As the first try on the quantitative evaluation of the disentanglement performance in the graph domain, the proposed evaluation protocol is interesting and deserves attention from a larger audience. But I have one concern about the computation of the $GED\_E$. As mentioned in the main paper, computing the full GED between the predicted factor graphs and the ground truth can be too expensive. So $GED\_E$ assumes that the order of nodes is fixed and correct. This would be a problem at least for the synthetic dataset since mixing the factor graphs together may generate some new factor graphs that are not included in the ground truth. Correct me if I am wrong. 2)In Tab. 1, it is great to have the random method to show the lower bound of the evaluation metric, but why only the $GED\_E$ is reported? The std of the random method should also be reported. 3)It would be good to have a visualization of the generated features from the DisenGCN.

Correctness: The claims of the proposed method are evaluated empirically on several datasets include synthetic and real-world ones. The empirical methodology to conduct graph-level disentanglement is reasonable.

Clarity: This paper is well written. The intuition of the proposed method is clearly conveyed. The description of the main method is concise.

Relation to Prior Work: The prior works have been discussed thoroughly and the difference of the proposed method from other methods is clearly discussed

Reproducibility: Yes

Additional Feedback: Please address my concerns in the Weakness Section. Some Typos, Line30, CNN-bsed --> CNN-based line 45, real-word --> real-world line 152 diverge --> diverse line 157 take palce --> take place line 187 sythetic --> synthetic After-rebuttal I have read the responses and other reviewers' comments. The authors have addressed most cocerns, so I keep the original score.


Review 2

Summary and Contributions: The submission proposes a new architecture of graph neural networks. In this new architecture, the model factorizes the graph into several graphs and then put GNNs over these graphs, then the learned representations are concatenated to get the final representation. Comparing to previous methods, which need graph factors beforehand, this method autometically learns factor graphs from the input graph. The empirical evaluation indicates that the proposed algorithm improves over several baselines.

Strengths: The strengh of the proposed method is the learnable factorization. This design does not require manually designed graph factors. Furthermore, the graph factor learning also have chances to improve the model performance. The work is very relevant to the community.

Weaknesses: The expriment of the proposed method is weak. The comparison does not include recent methods on graph classification. The performance improvements are marginal on real datasets. Does the method work for node classification? If so, then there should be a comparison with [Liu et al. 2019].

Correctness: The performance comparsions have some flaws. In table 2, the "2" column, all performance numbers are the same but only the proposed method are in bold. It seems that performance values are bold according to the mean. Do you do paired t-test to compare different algorithms? The submission claims that the proposed model are able to disentangle graph factors. However, the train procedure encourages the reduction of training error as well as the diversity of factors. Why do these factors need to be consistent with the groundtruth factors?

Clarity: The paper is hard to understand. I expect to see some high-level formulation of the problem and the motivation behind, but the method section starts with detailed introduction of the architecture. Some notations are not defined. For example, what's readout and f in (2)? What's the pooling method in graph classification?

Relation to Prior Work: Prior work are discussed.

Reproducibility: No

Additional Feedback:


Review 3

Summary and Contributions: This paper propose a new graph neural network Factor GCN which first explicitly construct several factor graphs based attention-like mechanism, then perform feature aggregation on each factor graphs separately. The main contribution of the paper is (i) a new GNN framework that can learn disentangled graphs; and (ii) experimental results show the effectiveness of the proposed method for learning disentangled graphs and graph classification

Strengths: 1. The idea is interesting and novel 2. Experimental results show the effectiveness of the proposed method for learning disentangled graphs and graph classification 3. The paper is well written and easy to understand

Weaknesses: 1. The proposed framework is very similar to GAT, except some minor difference in the attention-mechanism and the discriminator. The authors are encouraged to add the discriminator to GAT to show if the designed approach for constructing factor graphs is really more effective than GAT. 2. Some details are missing. For example, in figure 3, are the correlation analysis by averaging all the graphs or only on one graph? In Fig. 4, it is unclear on which dataset are the experiments conducted. When varying lambda,what is the number of factors and when varying number of factors, what is the value of labmda.

Correctness: Yes

Clarity: Generally, the experimental is well written except that some details are missing.

Relation to Prior Work: Yes

Reproducibility: Yes

Additional Feedback: Thanks for the rebuttal.


Review 4

Summary and Contributions: The goal of paper is to find disentangled representation for nodes of a graph. The method named FactorGCN has three steps: 1) Disentangling: Extracting disentangled factor graphs (which are independent as much as possible) 2) Aggregation: Aggregating features of nodes in each factor graph by utilizing GCN layers 3) Merging: Concatenation of features of one node in different factor graphs Its goal is providing low-dimensional features for nodes of the graph that are disentangled. The final goal is to have better results in the graph analysis tasks (like graph classification).

Strengths: The authors have worked on the experiments to show that the effectiveness of the proposed method. They have also proposed two metrics for evaluating how good the factor graphs are. The introduced synthetic dataset which makes us capable to compare factor graphs with the ground truth ones is also interesting.

Weaknesses: 1) The novelty is limited. 2) Although the paper starts with the claim that their main focus is on the multi-relation disentangling, there is nothing explicit in the method that emphasizes on the multi-relational graphs. It's not also provided explicitly in the experiment part. Can the proposed method give a heterogenous or multi-relational network (for which the type of edges is available and thus can be utilized for evaluation) and distinguish edges of different types by factorizing? 3) It seems that the graph classification loss may not provide enough supervisory signal for disentanglement. Why does enforcing classification of factor graphs make them disentangled necessarily?

Correctness: The authors’ claims are supported by extensive experimental results.

Clarity: The paper is not well written. Some parts of the paper is not clear. It's not easy to understand the final goal of the paper and what is the superiority of the proposed method to the previous work.

Relation to Prior Work: As said before, the paper starts with the claim that the main focus is on the multi-relational graphs, but none of the previous methods that focus on the multi-relational data is mentioned. It's not clear that the focus of paper is on what task in graph analysis. If the method is not task-specific, it should be tested on different tasks and prove the claim.

Reproducibility: No

Additional Feedback: 1) In Figure 2, the disentangled factor graphs have structures with different number of nodes that is not compatible with the description of method (i.e. "we first convert the continue value in the factor graph to 1/0 value by setting the threshold to make the number of edges in this two graphs are the same"). 2) Why did the authors focus on graph classification and not on node classification? There is nothing specific to graph classification, on the other hand, there are some ambiguity in the graph classification. 3) Why hasn’t the proposed method been compared with IPDGN? 4) What are the input features for each dataset, if any? 5) You should compare your method with some graph-classification methods, not just methods for node classification like GCN. 6) Isn't a sigma on e in the right-hand-side of Eq. (3) required? 7) There are many typos in the paper like: - CNN-bsed -> CNN-based - continue value -> continuous value - this two graphs -> these two graphs - it can in fact performance well -> it can in fact perform well - only based the its structure - rate of 0.002 No weight -> rate of 0.002. No weight...

[Author Response · NeurIPS 2020]

1. We sincerely thank the reviewers for the constructive comments and would like to address them as follows.

2. **{R1.1 Computing $GED_E$}** Yes, for the synthetic dataset, $GED_E$ may contain noises due to the reason mentioned by
3. R1. However, for real-world datasets like ZINC, $GED_E$ will be the same as GED, yet with a much lower complexity.
4. **{R1.2 Random method}** The $GED_E$ and C-Score are $32.09 \pm 4.85$ and $0.315 \pm 0.002$ when using four factor graphs.
5. **{R1.3 Visualization of DisenGCN}** We add the correlation visualization of DisenGCN, shown in the figure below.

6. **{R2.1 Graph classification methods}** Thanks for the comment. In fact, we have indeed
7. compared with GIN, published in 2019 and the SOTA graph-classification method on the
8. datasets we used (10-fold c.v.). As suggested, we have also added results of DiffPool and
9. GatedGCN, both tailored for graph-level tasks. On the ZINC dataset, the MAEs for DiffPool,
10. GatedGCN, GIN, and Ours (FactorGCN) are 0.466, 0.437, 0.387, and 0.366, showing the
11. superiority of our method, let alone its capability for graph-level disentanglement.

12. **{R2.2 Improvement}** The performance improvement is truly not trivial. On ZINC, a large-
13. scale real-world dataset, FactorGCN achieves an MAE of 0.366, while GIN, MoNet, GAT achieve 0.387, 0.397, 0.479
14. under the same setup [1]. The improvement over the best model is 5%, which, given the nature of the task and SOTA
15. results, can be indeed considered as significant. Such claim is also supported by the paired t-test.

16. **{R2.3 Node classification and IPDGN(Liu et al. 2019)}** We add node-classification experiments on the large "Pattern"
17. dataset (14K graphs, 1,664,491 nodes) [1]. Results including IPDGN are shown below. All models contain four layers.

18. **Method (Acc)** | Random (50.0)   GCN (63.9)   GatedGCN (84.5)   GIN (85.6)   MoNet (85.5)   DisenGCN (75.0)   IPDGN (78.7)   Ours (**86.6**)

19. **{R2.4 Paired t-test}** As suggested, we conduct paired t-test between our method and the second-best ones. The p values
20. on the Synthetic, ZINC, IMDB-B, COLLAB, and MUTUG are $< 0.0005$, $< 0.0005$, $> 0.25$, $> 0.25$, and $> 0.25$. It
21. shows that our method performs significantly better on the first two datasets and on par with SOTA for the rest.

22. **{R2.5 Consistency of factor graphs}** Thanks. Like any other disentanglement method [3], there is no absolute guaranty
23. that the disentangled factors will be strictly consistent with the ground truth. However, by enforcing the diversity of
24. factors, as also done in many disentanglement methods like [3], the model tends to generate factors that contain the
25. natural patterns (ground-truth ones) of the input. This is empirically supported and validated by the higher $GED_E$.

26. **{R2.6 High-level formulation & motivation}** Thanks for the nice suggestion. The core idea can be formulated as
27. $\arg\max_{\mathcal{F},\mathcal{D}} \big(\mathbf{P}(\mathbb{G} \subseteq \mathcal{F}(G), \mathcal{D}(\mathcal{F}(G)) = \mathbb{Y}|G)\big)$, where $\mathcal{F}$ will disentangle $G$ to a set a factor graphs, and $\mathcal{D}$ will
28. generate the label of $G$ based on the factor graphs. We seek the optimal $\mathcal{F}$ and $\mathcal{D}$ to maximize the probability that the
29. generated factor graphs contain the ground truth ones and the predicted label equals to the true label. Our motivations
30. are two-fold: disentangling the input globally will account for higher-order relations among nodes; multi-relational
31. disentangling will allow us to discover various relations between the same pair of nodes. We will add these to revision.

32. **{R2.7 Clarity}** Thanks. We will remove bold fonts in Tab. 2. Readout in Eq. 2 and pooling method are mean pooling.

33. **{R3.1 Modified GAT}** Thanks for the nice suggestion. As advised, we add a discriminator to GAT; the Micro-F1($\uparrow$),
34. $GED_E(\downarrow)$, and C-Score($\uparrow$) on synthetic dataset are 0.928, 12.35, 0.274, while those of ours are 0.995, 10.56, and 0.532.
35. It shows that the factor-graph method indeed leads to better disentanglement performances.

36. **{R3.2 More details}** Thanks. The correlation is computed on all graphs. Fig.4 is conducted on the ZINC dataset. When
37. varying $\lambda$, #factors is set to be eight; when varying #factors, $\lambda$ is set to be 0.2. We will add the details to the revision.

38. **{R4.1 Novelty}** We kindly solicit R4, if possible, to re-assess the novelty from task- and evaluation-perspective. Task-
39. wise, we introduce the first graph-level disentanglement method via GNN; metric-wise, for the first time we propose a
40. quantitative evaluation protocol of the graph disentanglement. Both could potentially interest a large audience in GNN.

41. **{R4.2 Multi-relational methods}** In fact, the setup of disentanglement is different from that of multi-relational models:
42. the former one aims to factorize a *single-relational* input graph into multiple graphs, while the latter requires a *multi-*
43. *relational* graph as input. As a result, the former task is expected to be much challenging than the latter. Due to the
44. setup difference, the proposed method does not support taking heterogeneous network as input. We will clarify this.

45. **{R4.3 Enforcing classification}** Thanks. The classifier will prevent model from collapsing to the point where all the
46. factor graphs are the same (e.g. all equal to the input), and encourage different factor graph to focus on different
47. sub-structures of the input. Without the classifier (i.e., setting $\lambda = 0$), the $GED_E$ will degrade from 12.6 to 13.0.

48. **{R4.4 Clarity: goal and superiority}** Our method aims to disentangle an input simple graph into several factor graphs
49. via a GNN. For the first time, this is done via a graph-level factorization and quantitatively evaluated using the proposed
50. graph-disentanglement metric, let alone the SOTA results. As suggested, we will clarify this in the revision.

51. **{R4.5 Additional feedback/Relation to prior work}** **[#node in Fig. 2]** Isolated nodes in Fig. 2 are removed for better
52. visualization. The #node of each graph is in fact the same. **[Tested on different tasks]** The manuscript includes graph
53. regression and classification tasks. We also add node classification task (please refer to R2.3), where **IPDGN** is added.
54. **[Input]** For synthetic, IMDB-B, and COLLAB, uniform embedding is used as the node feature; for ZINC, each node
55. has an one-hot vector representing the atom type (28 in total); for MUTAG, seven bios are used as node features.
56. **[Graph-classification methods]** Thanks. Please refer to R2.1. **[$\sigma$ in Eq. 3]** Yes, $G_i^e[c]$ in Eq. 3 contains an activation
57. function ($\sigma$) implicitly. **[Typos]** Thanks and we will fix them.

58. **[1]** Dwivedi, Prakash, et al. Benchmarking graph neural networks. **[2]** Xu, Keyulu, et al. How powerful are graph neural networks?
59. **[3]** Locatello, Francesco, et al. Challenging common assumptions in the unsupervised learning of disentangled representations


[Meta-Review · NeurIPS 2020]

Taking into account the reviews, rebuttal and the discussion, I think the paper is of interest to the community and I believe the empirical results are significant. However, I *urge* the authors to consider all the feedback given, and incorporate the points made in the rebuttal into the main paper. It is crucial for it to have the impact it deserves in the community. Ignoring it will minimize the impact the paper will have. So please take this chance and make sure the camera ready paper is as strong as it can be.